# Casirivimab-imdevimab monoclonal antibody treatment for an immunocompromised patient with persistent SARS-CoV-2 infection: a case report
Karun Saathveeg Sam, Pooja Khosla, Vinus Taneja ⑩ ✉ & Rishikesh Dessai

## Abstract

**Background** Persistent acute SARS-CoV- 2 infection is characterised by the persistence of symptoms of a COVID-19 illness and a persistently positive SARS-CoV-2 RT PCR result. It is often seen in immunocompromised individuals. Currently, there are no approved treatment regimens or guidelines for management.
**Methods** Our patient is a middle-aged male who had received chemotherapy prior to the onset of his SARS- CoV-2 infection and subsequently was diagnosed with a persistent and acute SARS- CoV-2 infection after presenting to us with a pyrexia of unknown origin. He was treated on an off-label basis following informed consent with casirivimab-imdevimab monoclonal antibody therapy, comprising two IgG1 neutralising human monoclonal antibodies at a dose of 600 mg each.
**Results** Our patient had significant clinical improvement on treatment with the monoclonal antibody therapy and it was well tolerated without any adverse effects. He is currently doing well during follow up.
**Conclusions** Casirivimab-imdevimab monoclonal antibody therapy could be beneficial for people with persistent acute SARS-CoV-2 infection. Therapy is warranted on a case-to-case basis. This highlights the need to identify immunocompromised individuals who are at risk of developing persistent acute SARS-CoV-2 infection to enable their prompt treatment.

## Plain language summary

The COVID-19 pandemic was associated with significant mortality and morbidity globally. People with weakened immune systems, such as cancer patients undergoing chemotherapy, are at an increased risk of developing a prolonged course of COVID-19. At present, no effective treatment options are available to treat SARS-CoV-2 infection in such a group of patients. Here, we describe a patient with Hodgkin's lymphoma and papillary carcinoma of the thyroid who went on to develop persistent COVID-19 and made a full recovery following treatment with anti-SARS-CoV-2 monoclonal antibodies. These results highlight the importance of considering similar treatment options for persistent COVID-19 in cancer patients undergoing chemotherapy, upon appropriate clinical evaluation.

Post-acute sequelae of SARS-CoV-2 infection have emerged as a progressively prevalent disorder significantly impacting individuals' quality of life. Approximately 10% of individuals who contract the COVID-19 virus are estimated to experience post-acute COVID-19 syndrome. Immunocompromised patients face an elevated risk of this condition due to their inability to mount an adequate immune response during the initial infection, also placing them at risk of a state where recovery from the acute illness is impeded by persistent viral shedding. Another clinically significant entity known as persistent acute COVID state or ongoing viral replication is seen in this subset of patients. Presently, there are no approved treatment regimens or guidelines for this condition. Given its diverse manifestations, it is essential to recognise this syndrome and explore potential treatment options that may benefit different patient cohorts. We share the case of a middle-aged male who underwent chemotherapy before contracting SARS-CoV-2 and subsequently was diagnosed with a persistent and acute SARS-CoV-2 infection. His swift clinical response to monoclonal antibody therapy underscores the importance of considering such interventions on a case-by-case basis.

Department of internal Medicine, Sir Ganga Ram Hospital, New Delhi, India. ✉e-mail: card.neuro@gmail.com

## Methods

A 58-year-old gentleman was diagnosed with Hodgkin's lymphoma and papillary carcinoma of the thyroid in 2021 and was started on the ABV-D(Adriamycin, Bleomycin, Vincristine and Doxorubicin) regimen as indicated. His response to treatment was satisfactory. He developed an insidious onset low-grade fever with cough and expectoration two months following the third cycle of chemotherapy and was hospitalised. His SARS-CoV-2 RT PCR done on a nasopharyngeal swab was positive. The SARS CoV 2 RT PCR was a real-time PCR which detected the following PCR sequences: N gene at 530 nm using a FAM-ACCCCGCATTACG TTTGGTGGACC probe, RdRp gene at 670 nm using a FAM-CAGGTGGAACCTCATCAGGAGATGC, E gene at 530 nm using a FAM- AAGGTTTTACAAGACTCACG with control at 560 nm. He was diagnosed with Severe SARS-CoV-2 pneumonia and managed successfully in the intensive care unit. Following discharge, he continued to have symptoms of fatigue and myalgia for over two weeks. He progressively went on to develop a low-grade fever which was intermittent, not associated with chills or rigours, not associated with rash, genitourinary, gastrointestinal or neurological symptoms. Three weeks later, he developed an insidious onset dry cough which was intermittent and not associated with chest pain or dyspnoea. These symptoms persisted beyond 3 months and his SARS-CoV-2 RT PCR remained positive on multiple occasions over the course of this illness. A local physician treated him for his symptoms which failed to resolve even after a course of oral antibiotics. There was no history of travel, no contact with pets, no history of contact with a person infected with tuberculosis, no history of prolonged antibiotic use or high-risk sexual behaviour.

At the time of the current presentation, he was alert, oriented and febrile with a temperature of 99.8 F. His pulse rate was 98 beats per minute, blood pressure was 110/70 mm Hg and oxygen saturation was 98% on room air. There were no rashes, generalised lymphadenopathy, oral ulcers, genital ulcers, nail or eye changes. On systemic examination, respiratory system examination revealed bilateral expiratory coarse crepitations in the anterior lung fields without any other added sounds. Other system examinations was unremarkable

Owing to the persistence of fever and cough for over 3 months, the history of a COVID-19 infection before the same and the persistently positive SARS-CoV-2 RT PCR, a persistent acute COVID syndrome was considered as a strong differential diagnosis. His clinical presentation was similar to that seen in people with long COVID syndromes however, he had a persistently positive RT PCR, whilst people with long COVID syndromes are noted to have negative RT PCR after some time of 2 to 4 weeks. A relapse of the pre-existing malignancy or metastases was also a strong possibility. Instances of autoimmunity following SARS-CoV-2 infections had also been reported and was another differential which was lower down on our list.

Upon evaluation, his baseline parameters were within normal limits (Table 1). His inflammatory markers were elevated at the time of presentation: C reactive protein of 118 MG/L(normal range 0-5 MG/L), IL6 199 ng/ml (normal range 7–18 ng/ml), Ferritin 704 ng/ml (normal range 25–350 ng/ml) and D Dimer 0.6 ng/ml (normal range: < 0.5). 12 lead ECG, Chest x-ray, Ultrasound abdomen and 2D ECHO were normal. An extensive infective workup was done which was unremarkable: Urine routine was normal, urine and blood cultures were sterile, and induced sputum samples for gram stain. Culture, AFB stain, TB gene Xpert, Kinyoun stain for Nocardia and KOH mount were negative; EBV and CMV PCR was negative; Dengue, rickettsial and leptospirosis serology was non-reactive; Malarial antigen test and peripheral smear was negative; HIV antigen and antibody testing was negative; HbsAg and Anti HCV were also negative. Serum procalcitonin was normal. Repeat clinical examination done on multiple occasions was unremarkable.

A whole-body PET CT was performed which showed bilateral lung lesions with mild hypermetabolic mediastinal lymph nodes. Bronchoscopy was performed which was normal. BAL analysis was done which was positive for the SARS-CoV-2 RT PCR. BAL gram stain, culture, KOH mount, galactomannan, Kinyoun stain, AFB stain and TB Gene Xpert were negative. Owing to persistent fever, an autoimmune workup consisting of Antinuclear antibody (ANA) by indirect immunofluorescence, ANA profile, C and P Anti neutrophil cytoplasmic antibody (ANCA) and complement levels was done, which was negative.

Respiratory multiplex PCR panel from the nasopharynx remained positive for SARS-CoV-2 on Days 1, 4 and 7 of admission and was notably positive before hospitalisation for 3 months. His covid 19 total antibody test and IgG levels were low despite the persistence of symptoms and a positive RT PCR test following a symptomatic primary infection three months ago. Having ruled out other potential causes for the dry cough, and with the available evidence, he was diagnosed as being in a state of persistent acute SARS-CoV-2 infection.

Following the diagnosis, there was a dilemma in the modality of treatment to pursue owing to the lack of guidelines and evidence. He was initially on corticosteroids with broad-spectrum antibiotic cover but the fever persisted. Since there was no evidence of infection and taking into consideration that our patient could not mount a sufficient immune response as indicated by his extremely low covid antibody titres, it was decided to administer the Casirivimab- Imdevimab monoclonal antibody cocktail, The antibody cocktail consists of two IgG1 neutralising human monoclonal antibodies. Casirivimab, a human immunoglobulin G-1 (IgG1) monoclonal antibody (mAb), is a covalent heterotetramer consisting of 2 heavy chains and 2 light chains produced by recombinant DNA technology in Chinese hamster ovary (CHO) cell suspension culture and has an approximate molecular weight of 145.23 kDa. Imdevimab, a human IgG1 mAb, is a covalent heterotetramer consisting of 2 heavy chains and 2 light chains produced by recombinant DNA technology in Chinese hamster ovary (CHO) cell suspension culture and has an approximate molecular weight of 144.14 kDa. Each drug was available at a concentration of 300 mg/2.5 mL (120 mg/mL) after dilution. 10 ml of the solution contained 600 mg of each antibody.

The risks involved were explained in detail to the patient and his family and written informed consent was obtained from the patient and the patient's family. The informed consent form clearly mentioned that the patient's data might be used for publication and/or educational purposes. Permission was also obtained from the medical superintendent and the Ethics committee of Sir Ganga Ram Hospital, Delhi where the patient was hospitalised.

### Table 1 | Baseline investigations

| Test | Result |
|---|---|
| Haemoglobin | 11.9 g/dl |
| Total leucocyte count | 8.80 thou/µl |
| Differential count | Neutrophils 50%, Lymphocytes 36%, Monocytes 10, Eosinophils 4% |
| Platelets | 2,19,000/µl |
| Blood urea nitrogen | 10.43 mg/dl |
| Creatinine | 0.86 mg/dl |
| Sodium | 142 mEq/l |
| Potassium | 3.98 mEq/l |
| Total bilirubin | 0.9 mg/dl |
| Direct bilirubin | 0.4 mg/dl |
| AST/ALT | 18/19 IU/L |
| ALP/GGT | 90/34 IU/L |

Abbreviations: *AST* aspartate aminotransferase; *ALT* alanine aminotransferase; *ALP* Alkaline phosphatase; *GGT* Gamma-glutamyl transferase.
SI conversion factors: To convert leukocyte count to 109, multiply by 0.001; haemoglobin to grams per litre, multiply by 10; platelets to 109 /L, multiply by 1.0; urea to millimoles per litre, multiply by 0.357; creatinine to millimoles per litre by 88.4; sodium and potassium to millimoles per litre, multiply by 1; bilirubin to micromoles per litre, multiply by 17.104; ALT, AST to microkatals per litre, multiply by 0.167.

## Reporting summary

Further information on research design is available in the Nature Portfolio Reporting Summary linked to this article.

## Results

The patient responded rapidly to the antibody cocktail therapy with a resolution of fever within 24 hours and cough within 3 days without any adverse effects. His inflammatory markers showed a falling trend (CRP 30 MG/l, IL-6 1.8 ng/dl, Ferritin 615 ng/ml, D Dimer 0.2 ng/dl) on the 6th day after receiving the antibody cocktail. His repeat SARS-CoV-2 19 RT PCR was negative on the 7th day after administration of the antibody cocktail. He was discharged on a tapering dose of steroids and anticoagulation as indicated by his baseline comorbidities. On follow-up, the patient was doing well with no relapse of symptoms and underwent his next cycle of chemotherapy uneventfully. Consent for the publication of his clinical details was obtained from the patient during follow-up.

## Discussion

Persistent acute SARS-CoV-2 infection is a lesser-known sequelae of SARS-CoV-2 which is being increasingly recognised in the immunocompromised. It is characterised by the persistence of symptoms of a COVID-19 illness and a persistently positive SARS-CoV-2 RT PCR[1]. This entity is often a diagnosis of exclusion in a patient presenting with a pyrexia of unknown origin. It is prudent to differentiate this entity from the other sequelae of acute COVID-19. Notably, long-term COVID or post-acute sequelae of COVID-19 is another debilitating condition with multisystem involvement that has been increasingly identified. The estimated 10% incidence among those infected is considered to be significantly lower than the true numbers. Long COVID is difficult to diagnose owing to its varied clinical spectrum ranging from non-specific complaints such as fatigue, lethargy, anhedonia to organ-specific manifestations to the tune of cognitive dysfunction, autonomic dysfunction, persistent cough, dyspnoea, coagulopathy, new-onset diabetes and the like[2]. The diagnosis of long-term COVID-19 is based on the persistence of symptoms in the absence of an antigen positivity for SARS-CoV-2.

The presence of similar symptoms with the evidence of viral shedding would suggest a persistence of the acute illness rather than a post-acute COVID syndrome or long COVID. In patients with depleted B cell function due to the underlying disease or chemotherapy, an entity termed as 'Depletion associated prolonged covid 19 ' has recently been coined[3]. This is to primarily differentiate these patients from long-term COVID-19 and to also simultaneously attribute the persistent viral shedding to the underlying B cell depletion. Also, this entity is specific to a group of patients who might require intensive treatment options as compared to other immunosuppressive states.

Though there is a lack of validated therapeutic options, reports of various seemingly effective treatment strategies have been noted. There are a large number of currently ongoing trials on different modalities of intervention for the long COVID such as pharmacotherapy, rehabilitation, alternative medicine, psychotherapy and patient education. Notably, the drugs being studied include corticosteroids, anticoagulants such as apixaban, anti-fibrotic, erythropoietin, monoclonal antibodies and various small molecules among others[4].

The role of monoclonal antibody therapy for persistent acute COVID has not been substantiated by randomised control trials. Ballotta et al. in reported the first documented use of casirivimab and imdevimab in a patient with persistent SARS-CoV-2 infection with an underlying haematological malignancy[5]. Effective clearance of persistent COVID has been documented with the same drug in two other instances[3,6]. Casirivimab and imdevimab are human immunoglobulin G-1 (IgG1) monoclonal antibodies that bind non-competitively to the non-overlapping domain of the SARS-CoV-2 spike protein blocking attachment to the human ACE2 receptor. The drug has been authorised for use among patients who are not vaccinated or are immunocompromised within the first 10 days of onset of symptoms. It is conditionally recommended in those with moderate disease provided there are no detectable covid antibodies[7]. However, following the emergence of the omicron variant, the antibody cocktail has not been efficacious and is currently not recommended[8].

Our patient unlike the above-mentioned reports was not on B cell depleting chemotherapy. He was on the ABVD regimen at the time of the initial diagnosis of acute COVID-19 pneumonia. However, owing to his underlying B cell malignancy, abnormal lymphocyte function and a subsequent qualitative B cell dysfunction could be postulated. Owing to the extent of his immunosuppression he was unable to mount sufficient antibody response as evidenced by his low IgG titres leading to the persistence of the virus, whilst otherwise exhibiting similar symptoms to people living with long covid. Considering the persistence of fever accompanied with a positive SARS-CoV-2 RT PCR during this time and the clear lack of another plausible explanation for his symptoms, off-label use of casirivimab-imdevimab was administered. Remarkably his symptoms subsided with 72 hours and his SARS-CoV-2 RT PCR was negative after 7 days of initiating the drug.

Anecdotal reports of similar benefits occurring among a similar group of patients in other centres with monoclonal antibody use prompted us to share our knowledge and clinical experience in treating our patient. Monoclonal antibodies might have a role that extends beyond the initial phase of acute illness. Long covid and persistent acute COVID remain a burden for patients and with no effective treatment, it is worthwhile to consider such therapy after documenting the lack of antibody response. Randomised control trials are needed to recommend the drug among immunocompromised patients with haematological malignancies. To our knowledge, this is the first case report of casirivimab-imdevimab use in a patient suffering from persistent acute COVID-19 on non-rituximab-based chemotherapy.

## Data availability

All data supporting the findings of this study are available within the paper.

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

## Author contributions

K.S.S. and V.T. prepared the manuscript. K.S.S., V.T., P.K., and R.D. reviewed the manuscript.

## Competing interests

The authors declare no competing interests.
