## [Peer Review File · Communications Medicine]

Reviewers' comments:

Reviewer #1 (Remarks to the Author):

Major claims of the paper: The authors present an important case report of a patient on ABVD for Hodgkin's lymphoma, followed by persistent COVID as evidenced by ongoing detection of SARS CoV-2 RNA by SARS CoV-2 PCR.

Recommendations

--- the authors refer to this patient as having long COVID. The patient is a very interesting case but does not have long COVID (ongoing or new symptoms following acute illness), but, rather, a prolongation of his acute COVID. These are two distinct entities. One term that has been coined for this group of patients is "depletion associated prolonged (acute) COVID". Authors argue that this patient is not on B cell depleting therapy however he did have lymphoma indicating potential for abnormal lymphocyte function, which is worth mentioning. I would absolutely not refer to this case as long COVID though- it will create confusion in the literature. It is a case of persistent proven viral shedding and for this population of subjects who fail to clear initial infection, antibody therapeutics need further explanation.

It would be reasonable to keep the nicely written long COVID text, explaining that, in contrast to depletion-associated prolonged COVID, long COVID is a distinct entity where proof of ongoing respiratory virus shedding is lacking, so the indication for antibody therapy is less clear. (Reviewer notes having personally run 100 saliva and nasal qRT PCRs seeking to find SARS CoV-2 in these secretions in patients with long COVID symptoms, and virus was not detected in any samples, unpublished data).

More minor critiques:

--- need to indicate where the PCR came from, ? nasal, nasopharyngeal, or saliva

--- did anyone sequence this virus ? If so would indicate results, if not would recommend at least a brief acknowledgement of sequencing these depletion-associated-prolonged shed strains: some of these may ultimately mutate away from susceptibility to casirivemab, imdevimab

--- need to discuss the fact that casi, imdev is no longer approved for use for COVID at all in most places as omicron has mutated away and is not susceptible. The authors argument that we do need randomized clinical trials of antibody cocktails active against currently circulating strains is important, just need to clarify that the antibodies used in such trials should be adapted to current SARS CoV-2 strains.

With the above mentioned edits I do think it is important to get this paper published, further pushing the medical community to develop treatments for these patients as we have no antibody options (for anyone) at this time, and need to advocate that they be developed.

Reviewer #2 (Remarks to the Author):

Review of COMMSMED-23-0512-T

Overall: This is an interesting case report of a patient with immunocompromise and COVID-19

infection which went on for an extended period. I am not certain this constitutes “Long COVID” which usually has symptoms develop after the initial 30 day period post-diagnosis. Here, his symptoms just did not subside, although they lessened than to the initial degree. I would have called it more persistent acute COVID. I make this distinction because the approach to treatment would be different. Also, I think it’s important to differentiate between incident symptoms that define PASC, and persistence of symptoms that would be more extended acute COVID. The fact that the PCR test remained positive, also implies that this is not post-COVID but an extended period of acute COVID. That also can explain why the treatment with casirivimab-imdevimab monoclonal antibody was successful (but not to imply that it would not also work with PASC).

The authors should reference previous manuscripts that help define Long COVID and point out where this falls into that continuum. It was great to see the patient respond with this treatment, but again, not sure this is truly Long COVID.

Some specific comments:

1. This paper needs a medical writer/editor. There are many typos.
2. The PCR test is SARS-CoV-2 and not “COVID” test.
3. The authors note many lab values were out of normal range, but should give the cutoffs for the reader.
4. In table 1, the authors should include the leukocyte count differential, also.

Response to reviewers

Reviewer 1

1. Thank you for your response. Owing to the persistence of a positive SARS COV 2 rtPCR, our patient would fit more into a state of persistent acute covid and not long covid. We have made the changes to the manuscript along with the addition of relevant references.
2. We have mentioned clearly so as to specify the site of sampling for all the SARS COV 2 RT PCR samples in the manuscript.
3. No. Sequencing of the virus was not performed at our centre.
4. Yes. We have mentioned about the efficacy of the antibody cocktail towards the newer covid variants and that it is no longer recommended with references in the updated manuscript

Reviewer 2

1. Thank you for your response. Owing to the persistence of a positive SARS COV 2 rtPCR, our patient would fit more into a state of persistent acute covid and not long covid. We have made the changes to the manuscript along with the addition of relevant references. Further, we have included more information on long covid and on persistent acute covid in the discussion
2. We have reviewed the manuscript and corrected the typos.
3. The term 'SARS-COV -2 rtPCR' has been mentioned at all relevant places in the edited manuscript.
4. Normal ranges for the quantitative reports and changes suggested to the reports that have been tabulated have been carried out in the edited manuscript

REVIEWERS' COMMENTS:

Reviewer #1 (Remarks to the Author):

My comments have been fully addressed. I believe this interesting manuscript is now ready for publication.

Reviewer #2 (Remarks to the Author):

Thank you for your revisions and response to my critical review. I am satisfied you have successfully responded and have no further criticisms.